# Deep Learning of Cancer Stem Cell Morphology Using Conditional Generative Adversarial Networks

**DOI:** 10.3390/biom10060931

**Published:** 2020-06-19

**Authors:** Saori Aida, Junpei Okugawa, Serena Fujisaka, Tomonari Kasai, Hiroyuki Kameda, Tomoyasu Sugiyama

**Affiliations:** 1School of Computer Science, Tokyo University of Technology, 1401-1 Katakura-machi, Hachioji-shi, Tokyo 192-0982, Japan; saoaida@yamaguchi-u.ac.jp (S.A.); kameda@stf.teu.ac.jp (H.K.); 2Graduate School of Sciences and Technology for Innovation, Yamaguchi University, 2-16-1 Tokiwadai, Ube-shi, Yamaguchi 755-8611, Japan; 3School of Bioscience and Technology, Tokyo University of Technology, 1401-1 Katakura-machi, Hachioji-shi, Tokyo 192-0982, Japan; b01160559e@edu.teu.ac.jp (J.O.); b01162818c@edu.teu.ac.jp (S.F.); t-kasai@okayama-u.ac.jp (T.K.); 4Neutron Therapy Research Center, Okayama University, 2-5-1 Shikada-cho, Kita-ku, Okayama 700-8558, Japan

**Keywords:** Cancer stem cell, conditional generative adversarial network, phase contrast, green fluorescence protein, tumor

## Abstract

Deep-learning workflows of microscopic image analysis are sufficient for handling the contextual variations because they employ biological samples and have numerous tasks. The use of well-defined annotated images is important for the workflow. Cancer stem cells (CSCs) are identified by specific cell markers. These CSCs were extensively characterized by the stem cell (SC)-like gene expression and proliferation mechanisms for the development of tumors. In contrast, the morphological characterization remains elusive. This study aims to investigate the segmentation of CSCs in phase contrast imaging using conditional generative adversarial networks (CGAN). Artificial intelligence (AI) was trained using fluorescence images of the *Nanog*-Green fluorescence protein, the expression of which was maintained in CSCs, and the phase contrast images. The AI model segmented the CSC region in the phase contrast image of the CSC cultures and tumor model. By selecting images for training, several values for measuring segmentation quality increased. Moreover, nucleus fluorescence overlaid-phase contrast was effective for increasing the values. We show the possibility of mapping CSC morphology to the condition of undifferentiation using deep-learning CGAN workflows.

## 1. Introduction

Tumors are believed to be maintained by a minor population of cancer cells. These are termed cancer stem cells (CSCs) to describe the extraordinary characteristics of these cells provoking new tumors as determined by an allograft mouse tumor system [1]. The CSCs have the ability to grow themselves while maintaining an undifferentiated property and to generate progenitor cells with the potential to produce a major population of cancer cells. The first evidence of CSCs was reported in a study of blood tumor-initiating cells showing the hematopoietic stem cell (SC) surface marker, cluster of differentiation (CD), CD34^+^/CD38. Then, CSCs were isolated from solid tumors as the only cells capable of initiating new tumors. The cell-surface markers characteristic to CSCs were identified to separate them from other cells. For example, CD24^−/low^/CD44^+^, CD20^+^ in spheroid cells, and CD133^+^ were identified for human breast tumors, melanoma, and brain tumor tissues, respectively. Importantly, the CSC populations were extremely low in each tumor tissue. It is postulated that CSCs have an important role in chemoresistance and radiation resistance [2]. The development of new therapy according to the CSC concept is interesting, although the origin of the cells and their path to becoming CSCs remains unclear.

Cultured CSCs are useful as powerful tools of cancer research. It is appropriate to employ primary cultures of CSCs, which are selected using a fluorescent activated cell sorter with cell surface markers, regardless of whether these proteins are directly involved in the SC biology [2]. Another approach of CSC culture is the use of induced pluripotent stem (iPS)-derived CSCs [3]. Mouse iPS (miPS) cells have been shown to acquire characteristics of CSCs by treatment with the conditioned medium of cancer cell lines as a niche for SCs. Unlike normal miPS cells, they formed malignant tumor tissue after transplantation into nude mice. However, the stem-like cells taken from the tissue formed spheres on the attached culture and spheroids in the suspension culture which morphologically resembled miPS cells. These iPS-derived CSCs retained SC marker gene expressions such as *Nanog* and *Rex1* at even higher levels. *Nanog* and other genes have the capability of transforming normal cells into pluripotent cells [4]. The similarity between the reprogramming mechanisms of iPS cells by transcription factors and the mechanisms of cell transformation to CSC has been of interest [5].

It is widely accepted that SCs form morphologically typical colonies in the undifferentiated state [4,6]. Upon differentiation stimuli, SCs transform their shape, leading to several functionally distinct cell types. The CSC-like cells selected from human nasopharyngeal carcinoma cell lines exhibited distinct cell morphology from non-CSC-like cells [7]. Spherical colonies were formed by iPS-derived CSCs expressing *Nanog* but not by cells without the *Nanog* expression [3,8]. The tumor tissues developed from iPS-derived CSCs contained cells expressing both cancer cell marker protein and green fluorescent protein (GFP) reporting the *Nanog* expression. However, most cells expressed only one of these proteins, suggesting the production of differentiated cancer cells from iPS-derived CSCs to generate tumor tissues. Given these morphological characteristics of SCs, we hypothesized that CSCs in cultures might have typical cell morphology compared to cells that lost the SC marker gene expression.

Examination of cell morphology by phase contrast microscopy is a basic method for cell biologists to define cell shape and appearance based on basic categories such as fibroblastic, epithelial-like, and lymphoblast-like cells. Trainees with substantial cell culture expertise might detect signs of healthy cell status by inspecting the cells. It is not surprising that SC biologists may notice signs of deterioration of SCs and/or CSCs losing their pluripotent characteristics by checking the cells regularly. In recent years, image recognition technologies have made remarkable advances using artificial intelligence (AI). The methods have been applied for the identification of endothelial cells, as well as the classification of protein localization and cells [9,10,11]. An image-to-image translation system, called a conditional generative adversarial network (CGAN), is an advanced AI system wherein a photograph can be translated by a single algorithm without specific settings [12]. The examples of image translation are significant in terms of accuracy and creativity. The CGAN code learned a mapping between an input and output image. It is interesting to determine if the code can learn a mapping from a phase contrast image of an iPS-derived CSC to a GFP fluorescence image of the corresponding CSC. In other words, it is curious whether the code can recognize and distinguish the morphology of iPS-derived CSCs expressing GFP. We previously applied a deep-learning algorithm for the recognition of iPS-derived CSCs [13]. The AI accepted 10 221 pair images of cultured iPS-derived CSCs phase contrast and GFP fluorescent images to learn the cell morphology in relation to the *Nanog* expression. Although the mathematical formula of cell image recognition was unclear, the AI system displayed the capability of finding *Nanog*-expressing cells in phase contrast cell images after deep learning. Here, we examined the accuracy of output images by AI, which learned cell images taken under various conditions. We detected iPS-derived CSCs in phase contrast tumor tissue images by the deep learning of tissue image pairs.

## 2. Materials and Methods

### 2.1. Cell Culture

Lewis lung cancer (LLC) cells of mice, generously gifted by Dr. M. Seno (Department of Medical Bioengineering, Okayama University), were maintained in DMEM high glucose supplemented with 10% fetal bovine serum (FBS), 1× non-essential amino acids (NEA), and 1% penicillin/streptomycin (P/S) in a 5% CO_2_ incubator. These culture reagents were purchased from FUJIFILM Wako Pure Chemical Corporation, Osaka, Japan. A conditioned medium (*cm*) from LLC cells was collected from confluently grown cells in the same medium, except with 5% FBS for one day, and then filtered using a 0.45 μm filter. For a culture of miPS-LLC*cm* cells, a CSC model, multi-well plates, and dishes were pre-coated with 0.1% porcine-skin gelatin (MilliporeSigma, St. Louis, MO, USA) in a CO_2_ incubator for 12 h. The cells were maintained at 37 °C in a 5% CO_2_ incubator in a medium containing an LLC conditioned medium mixed with DMEM high glucose with 15% FBS, 1× NEA, 1% P/S, 1× L-glutamine (FUJIFILM Wako Pure Chemical Corporation), and 100-μM 2-mercaptoethanol in a ratio of 1:1 in accordance with a previous report [3]. Mitomycin C-treated mouse embryonic fibroblast (MEF) feeder cells (REPROCELL, Yokohama, Japan) were cultured in DMEM high glucose with 10% FBS, 1× NEA, and 1% P/S for three days before seeding the miPS-LLC*cm* cells on feeder cells.

### 2.2. Animals and Tumor Tissue Preparation

Cell cultures of miPS-T47D*cm* cells, a CSC model, and animal experiments were studied as previously described [3,8,14]. Briefly, four-week-old female Balb/c-nu/nu mice (Charles River, Yokohama, Japan) were subcutaneously injected with 7.5 × 10^5^ cells that were converted into CSCs with a conditioned medium treatment and suspended in 200 μL of phosphate-buffered saline (PBS). Tumors were harvested to a size of approximately 1000 mm^3^. Tumors fixed with 10% formalin were finally equivalated into 20% sucrose in PBS at pH 7.4, and embedded in an optimum cutting temperature compound (Sakura Finetek, Tokyo, Japan) at −80 °C. Five-micrometer-thick sections were cut and collected on glass slides.

The protocol for the animal experiments was reviewed and approved by the Animal Care and Use Committee of Okayama University under ID OKU2019591. All experiments were conducted in accordance with the Policy on the Care and Use of Laboratory Animals, Okayama University.

### 2.3. Microscopy

Living cells grown in multi-well plates were examined using a fluorescence microscope BZ-X800 (KEYENCE, Osaka, Japan) equipped with CFI Plan Fluor DL 10× (Nikon, Tokyo, Japan) and Plan Fluorite 20× LD PH objective lenses (KEYENCE).

Tumor sections on glass slides were incubated with 0.5-μg/mL Hoechst 33342 (Thermo Fisher Scientific, Waltham, MA, USA) in PBS for 15 min for nucleus staining. After washing the slides in PBS, the sections were mounted in PBS. The GFP fluorescence (525 nm) was visualized at a 470-nm excitation with an exposure time of 1 s. Hoechst 33342 fluorescence (460 nm) was visualized at a 360-nm excitation. Images were acquired as a set of phase contrast and GFP fluorescence images or as a set of phase contrast, GFP fluorescence, and Hoechst 33342 fluorescence images. All images were acquired at a resolution of 1920 × 1440 pixels and saved as tiff files. The CSC image diagnosis of the tumors was performed by a clinical technologist.

One sequentially acquired avian heart (purchased from a butcher) section was stained with a hematoxylin-eosin (HE) stain (MUTO Pure Chemicals, Tokyo, Japan) according to the manufacturer’s instructions. The other was stained with an Elastica van Gieson (EVG) stain (MUTO Pure Chemicals) for elastic tissues.

### 2.4. Image Processing and AI

For machine learning, the hardware was equipped with a Core i5-3470S CPU (Intel, Santa Clara, CA, USA), 32-GB PC3L-12800 memory (Kingston, Fountain Valley, CA, USA), and GeForce GTX1070Ti GPU (ELSA, Tokyo, Japan). For high-performance GPU-accelerated software environments, the NVIDIA CUDA Toolkit 8.0 (NVIDIA Corp., Santa Clara, CA, USA) was built on Ubuntu 16.04.1 LTS (Canonical Ltd., London, UK) with kernel version 4.4.0. The GPU-accelerated NVIDIA CUDA Deep Neural Network library (cuDNN) v6.0 (NVIDIA) was used for the deep-learning framework TensorFlow version 1.4.1 [15], which was built in Python 3 (https://www.python.org/). Each paired image file of phase contrast and fluorescence was divided into 35 files with a resolution of 256 × 256 pixels by a Python script utilizing the NumPy and PIL packages for Python. Each phase contrast image was joined with the corresponding fluorescence image to create a new image where the two images were arranged side by side. For CGAN software pix2pix port [12], a TensorFlow implementation was used according to practical instructions (https://github.com/affinelayer/pix2pix-tensorflow). The recall, precision, specificity, F-measure, and correlation coefficient values were used to evaluate the similarity between the output and the target. Precision is the fraction of the true positive cases that are actually positive among the predicted positive cases. The recall is the fraction of the true positive cases that are actually positive among the true positive and the false negative cases. The specificity is the fraction of true negative cases that are actually negative among the true negative and the false positive cases. Precision, recall, and specificity are defined by Equations (1)–(3), respectively.
(1)$$\mathrm{Precision}=\frac{TP}{TP+FP},$$
(2)$$\mathrm{Recall}=\frac{TP}{TP+FN},$$
(3)$$\mathrm{Specificity}=\frac{TN}{TN+FP},$$
where *TP* is the number of true positives, *TN* the number of true negatives, *FP* the number of false positives, and *FN* the number of false negatives. F-measure is a harmonic mean that combines both recall and precision. F-measure is defined by Equation (4),
(4)$$\mathrm{F-measure}=\frac{2RP}{R+P},$$
where *R* is recall and *P* is precision.

After binarizing the output and target images, the correlation coefficient between the two images was calculated. The correlation coefficient is defined by Equation (5):(5)$$\mathrm{Correlation}\;\mathrm{coefficient}=\frac{\sum_{m}\sum_{n}\left(F_{mn}-\mu_{F}\right)\left(G_{mn}-\mu_{G}\right)}{\sqrt{\left(\sum_{m}\sum_{n}{\left(F_{mn}-\mu_{F}\right)}^{2})(\sum_{m}\sum_{n}{\left(G_{mn}-\mu_{G}\right)}^{2}\right)}},$$
where *F* and *G* are the image area and *µF* and *µG* are the average values of *F* and *G*, respectively.

### 2.5. Statistical Analysis

Ryan’s method was used for the evaluation of the differences between groups. The Student’s *t*-test was used for the evaluation of the differences between two groups. Pearson’s chi-square test was used for the evaluation of independence of two categorical valuables.

## 3. Results

### 3.1. Deep Learning of CSC Image Cultured on Multi-Well Plate

We used miPS-LLC*cm* cells as a model of CSCs [3]. The *Nanog*-GFP reporter gene-harboring miPS-LLC*cm* cells allows us to easily acquire information on the pluripotency of cells by examining the GFP fluorescence [16]. The characteristics of miPS-LLC*cm* cells as CSC were previously proved by the evidence of the *Nanog* expression, diverse SC markers expression and the mouse in vivo experiments [3]. The SC markers were disappeared in correlation with the loss of the *Nanog* expression. The AI was expected to learn the morphology of miPS-LLC*cm* cells shown on phase contrast cell images in relation to the corresponding GFP fluorescence. We examined both the training and procurement of the AI that predicts GFP fluorescence positive miPS-LLC*cm* cells in phase contrast cell images without GFP fluorescence image information. Three types of image datasets were used for AI to evaluate the difference in magnitude of the objection lenses and the presence of MEF feeder cells (Figure 1a). The miPS-LLC*cm* cells on MEF had morphological characteristic features of dense, stacked, round, and aggregated cells, which differed from cells on the porcine-skin gelatin-coated surface. We observed that the GFP fluorescence of each cell did not show an equivalent intensity, although they were all GFP fluorescence positive. In fact, each cell within the same colony showed diverse intensity. The GFP fluorescence was almost absent in some cells. The fluorescence property was consistent with previous reports [3,16]. We utilized the software pix2pix to perform deep learning of cell images using CGAN [12]. Pix2pix accepts image pairs of phase contrast and fluorescence images (Figure 1b). The discriminator learns whether the image pair belongs to a real pair or a fake pair which includes images synthesized by the generator. The generator learns to trick the discriminator. Two hundred epochs were applied for all training.

Ten thousand cells per well were cultured in 96-well plates. A set of phase contrast and GFP fluorescence cell images was acquired at the center of each well using a 10× objection lens. The 96 sets of images were processed to obtain 3260 sets of 256 × 256 pixel images for AI training, and a hundred sets of those for the evaluation of the AI that was trained. We observed that the discriminator loss 1 value increased immediately after 7500 steps and reached a value of almost 1.3, suggesting that the discriminator failed to differentiate between real and fake GFP fluorescence (Figure 2a). The generator loss L1 value was lower at the end of training than the initial value. These changes in loss value were also observed in other trainings described as follows on the cultured CSCs. Next, we compared the AI-generated fluorescence image output with a paired image against the input as the target (Figure 2b). The output and target fluorescence images were not identical. The AI-generated fluorescence image in some cells was not present in the target. In other examples, AI did not draw fluorescence images in some cells where GFP fluorescence was observed. It is notable that AI never depicted fluorescence images in spaces where no cells were present.

Because the images used for the training included blanks with no cells, we eliminated these images for the next training. The training was performed with 2851 sets of images. However, we did not observe a marked improvement (Figure 2c). Next, to examine whether training was affected by the background gradient observed in the phase contrast image acquired using the 10× objection lens (Figure 1a), we eliminated all images other than the four pieces in the center of each image for training. The 300 sets of images were trained (Figure 2d). The similarities between outputs and targets (Figure 2d) improved slightly compared to the 100 outputs obtained by training with no selection of images (Figure 2b). We did not observe any depiction of fluorescence images from dishes coated with porcine-gelatin by AI models (data not shown).

Next, we examined training with 1526 sets of images acquired using the 20× objection lens (Figure 2e). The background of the phase contrast images was uniformly grayed compared to that with the 10× objection lens (Figure 1a). We observed detailed intracellular structures in cells from the phase contrast images; however, a robust improvement in output was not observed. Next, we examined miPS-LLC*cm* cells on MEF feeder cells in 24-well plates. The 3027 sets of images were trained (Figure 2f). Almost all aggregated colonies of miPS-LLC*cm* cells showed GFP fluorescence, although the intensity within the colony was not uniform. As shown by the outputs, AI did not miss drawing in the region of those colonies when never depicted in the region of MEF feeder cells. We did not observe any depiction of fluorescence images from dishes culturing MEF feeder cells by AI models (data not shown).

To evaluate the similarity between the output and target, we calculated the values of recall for true positive (Figure 3a), precision for false positive (Figure 3b), specificity for true negative (Figure 3c), F-measure for the weighted average of recall and precision (Figure 3d), and the two-dimensional (2D) correlation coefficient for image quality (Figure 3e). Interestingly, the training set with the 10× objection lens and center had significantly increased recall and precision values compared to the 10× objection lens. The maximum recall and precision values were from 0.80 to 1.0, although the mean values were from 0.16 to 0.55. By selecting images for training, the recall values significantly increased, whereas the precision values remained constant. The training set using MEF feeder cells showed the highest values of the training sets (Figure 3a,b). These observations were confirmed by the F-measure values (Figure 3d). In addition, we observed a similar training set effect on the 2-D correlation coefficient values (Figure 3e). In contrast, the mean specificity values were almost 1.0 for all training sets (Figure 3c), indicating that AI did not depict images where cells without GFP fluorescence were cultured and no cells were present.

### 3.2. Deep Learning of CSC Images in Tumor Tissue

To examine whether tissue with sequential sections was suitable for the training image set, we prepared HE- and EVG-stained tissues as a model. The training was performed to map from HE- to EVG-stain images using the 92 sets of pairs. All output images were different from their respective targets (data not shown). For example, some tissues were depicted in a region where no tissue was present. The position of the depicted wavy elastin was generally not true compared to the target image. Next, we prepared processed images in which additional coloring was drawn on the HE-stained images based on the corresponding EVG images. Using the processed image and the HE-stained image as a set for training, i.e., 140 sets for training, we obtained better outputs than those mentioned (data not shown). However, it was difficult to draw information precisely the same as the original EVG image information, such as the region and the color intensity. Thus, we conclude that it is difficult to prepare sets of tissue images for training using basic histological methods.

Then, we examined two sets of phase contrast and fluorescence images of tumor tissues derived from miPS-T47D*cm* cells, although phase contrast is not commonly used in pathophysiological study (Figure 4a,b). Characteristics of miPS-T47D*cm* cells as CSCs were previously proved by *Nanog* and diverse SC markers expression, and the mouse in vivo experiments [14]. It was reported that the disappearance of SC markers was correlated with the loss of the *Nanog* expression. The positive area for GFP fluorescence indicates the presence of *Nanog*-GFP reporter gene-harboring miPS-T47D*cm* cells which retained the CSC pluripotent characteristics. Consistent with the previous report [14], we observed randomly colonized GFP positive cells in glandular structures while all tumor cells were evenly distributed in the tumor. Although two loss function values suggest that the training using 2734 sets of phase contrast and GFP fluorescence images was not perfect (Figure 4c), it is surprising that there were examples of output drawing without color while the target had no GFP fluorescence (Figure 4d). Although the content in the 2734 sets differed, we did not observe a marked improvement in the training sets using the Hoechst 33342 overlaid-phase contrast instead of a simple phase contrast to create pairs with GFP (Figure 4e,f). As negative controls, we did not observe any depiction of fluorescence images from slide glass coated for tissue section (data not shown). By contrast, the classification of each set of 684 outputs indicates differences between the outputs (Table 1). Each output was diagnosed regardless of depiction. Then, the outputs were grouped into two types—those exhibiting GFP fluorescence and those not. We observed an increase in the ratio of depicting outputs to GFP fluorescence positive targets. The data was subjected to the Pearson’s chi-square test to see whether GFP depicting outputs were independent from the targets. We observed *p* < 0.01, indicating significant dependence of GFP-depicting outputs to GFP fluorescence positive targets.

Moreover, we evaluated the similarity between the output and the target using various values (Figure 5a–e). Interestingly, the training set with the Hoechst 33342 overlaid-phase contrast had a significantly higher recall value compared to sets without Hoechst 33342 (Figure 5a). Each maximum value was 1.0, although the mean values varied from 0.05 to 0.10. The training sets did not affect the precision values (Figure 5b). Accordingly, the F-measure value was significantly increased in the training set with the Hoechst 33342 overlaid-phase contrast (Figure 5d). A similar effect was observed in the 2-D correlation coefficient values (Figure 5e). In contrast, the mean specificity values were almost 1.0 for both training sets (Figure 5c). Although a significant difference was observed, it would be largely meaningless in view of tumor diagnosis.

## 4. Discussion

We applied CGAN to cell biology and performed CSC morphology learning for a novel method diagnosing the presence of *Nanog*-expressing cells in cultures and tumors. The development of AI demonstrated the capability of this approach. Not surprisingly, the accuracy of AI depended on the sets of images for learning, and had the potential to find CSCs in phase contrast images. For intensity optimization, exposure parameters while acquiring images needs to be seriously considered for the best AI model. Our results indicate that the AI developed in this study was not highly efficient in detecting *Nanog*-expressing cells compared to GFP fluorescence analysis; however, it could be improved for AI-aided diagnosis systems of CSCs.

New cytometrical methods have emerged from deep-learning technology to determine cell and tissue characteristics from images to understand the cell biology, physiology, and pathophysiology of samples [17]. Image segmentation is one of these important areas to be developed. Cell shape, nucleus, mitosis, and hemorrhage were automatically detected using convolutional neural networks (CNNs) [18]. U-Net, which requires a relatively small number of training data, efficiently acquired the segmentation of neuronal structure in tissues and cells in cultures [19]. For the segmentation of spheroids—a morphological shape often observed in SC suspension cultures [4]—the CGAN model was better than the U-Net model [20]. Fluorescent cell images were better segmented using CNN with an adversarial loss model than the CNN-only model [21]. Although these deep-learning workflows are efficient and sufficient for dealing with various types of images originating from conditions such as staining and brightness, the methods still require sets of images previously classified appropriately by experts in the field for datasets used in deep learning. It is obvious that these workflows are useful for known cytological structures. However, they may have difficulty indicating novel structures in cell and tissue images that have not been clearly defined by experts. In contrast, we used phase contrast images of CSCs, the morphological characteristics of which are clearly defined although CSC biologists might thereof have an empirical sense. In fact, the images contained CSCs and non-CSCs which seemed indistinguishable. The use of GFP fluorescence reporting *Nanog* expression was the only reliable way to distinguish between these cells. Thus, we used GFP fluorescence to define CSCs for training. Accordingly, the CGAN model was applied for the first time.

Interestingly, our results show the capability of AI to define structures not described clearly by experts. Cells that formed tube-like structures by the differentiation of miPS-LLC*cm* cells accompanied the loss in GFP fluorescence [22]. The GFP fluorescence was absent in fibroblast-like cells derived from miPS-LLC*cm* cells with morphological characteristics of a round shape, and high nuclear-to-cytoplasmic ratios [3]. Mouse embryonic SCs spread and became irregular when the *Nanog*-expression was diminished [23]. Compared to morphological changes in the literature, it was not easy to distinguish each *Nanog*-expressing miPS-LLC*cm* cell from cells that lost the expression in colonies. By contrast, the correlation values between the depicted and true images suggest that AI might detect morphological differences under phase contrast microscopy. Although the values of precision, recall, and F-measure were not efficient in our AI model compared to AI models generated by deep learning of known structures [18], the deep-learning workflows using CGAN could be improved by examining cell culture conditions and selecting images for training. Further studies would be required on the effect of the use of center images for the training set with 20× objection lens. The presence of MEF feeder cells showed the highest values of image evaluation. It is interesting to determine whether the increase in the values can be obtained using a training set with a selection of eliminating blanks and the center.

The CSCs in tumors have been identified by means of surface markers [8,24,25]. Identification of CSC markers accelerates the CSC concept [1,26]. The presence of CSCs in the hierarchical development of tumor tissue has been shown in many studies. By contrast, there have been limited descriptions of the CSC morphology in tumors. We observed that the AI model depicted CSCs in terms of GFP fluorescence using phase contrast images. The image qualities were not sufficient compared to that of the target; however, the improvement using the Hoechst 33342 overlaid-phase contrast suggests the morphological difference between CSCs and non-CSCs using microscopy. It could be interesting to investigate the mechanisms of the AI model in mapping phase contrast images to GFP fluorescence.

## 5. Conclusions

We investigated deep learning for the mapping of undefined CSC morphology. We used CGAN to generate AI models to segment CSCs in cultures and tumors. Segmentation of the CSC region was affected by the training set. The deep-learning framework using CGAN could be useful in identifying undescribed morphological characteristics in CSCs.

## Figures and Tables

**Figure 1 biomolecules-10-00931-f001:**
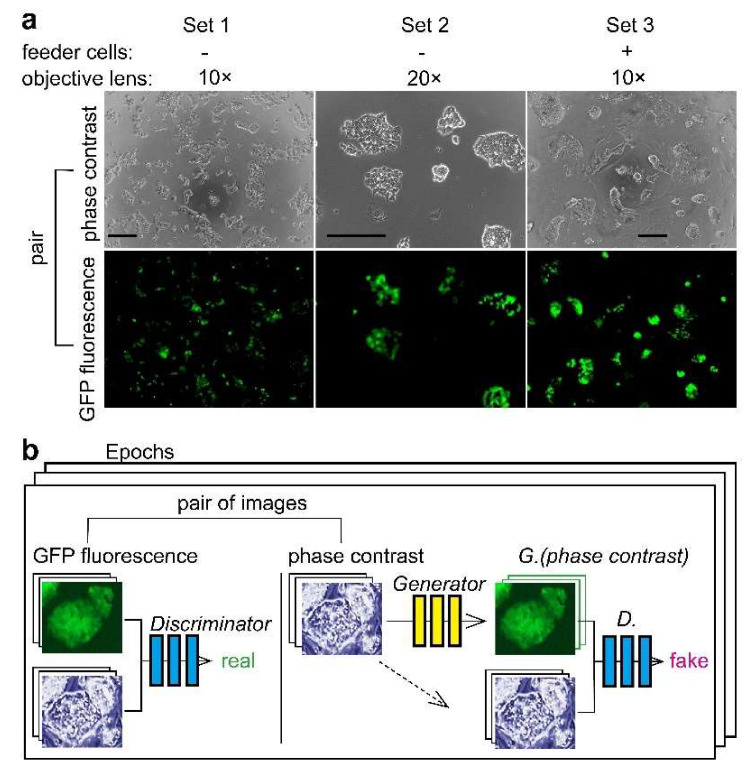
Experimental design of deep learning of miPS-LLC*cm* cell morphology. (**a**) Cells image sets for deep learning. For Sets 1 and 2, miPS-LLC*cm* cells were cultured in 96-well plates for 1–2 days. Cell images were taken using a 10× or 20× objection lens for each set. For Set 3, miPS-LLC*cm* cells were cultured for one day on MEF cells previously immobilized in 24-well plates. Bars = 200 μm. (**b**) Training a conditional generative adversarial network (CGAN) to map grayscale bright-field cell images into color dark-field fluorescence images. Learning was performed with several hundreds to thousands of images per epoch; the final epoch number was set to 200.

**Figure 2 biomolecules-10-00931-f002:**
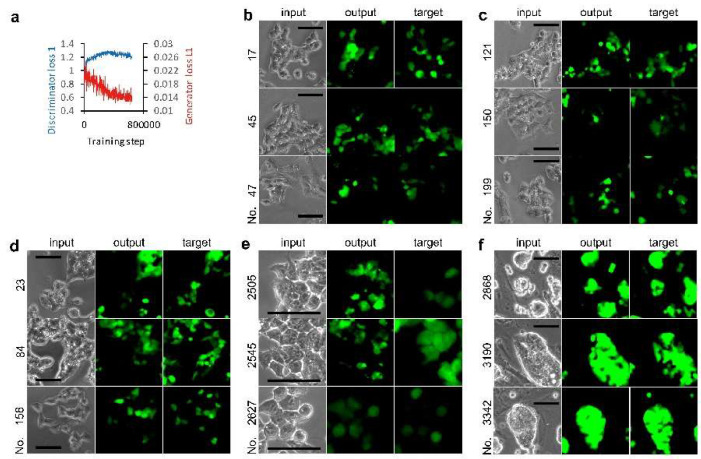
miPS-LLCcm cell image mapping from phase contrast to green fluorescent protein (GFP) fluorescence. (**a**) Effect of training steps on loss functions. (**b**–**f**) Output examples by AI models. Test phase contrast images were subjected to AI models for depicting fluorescence images. Input and target are the image of a pair for the evaluation of the depicted image. Images used for training AI for AI models: Set 1 images (**a**,**b**) without selection, (**c**) with selection of eliminating blanks, and (**d**) with selection of center; (**e**) Set 2 images with selection of eliminating blanks; and (**f**) Set 3 images with center selection. Bars = 100 µm.

**Figure 3 biomolecules-10-00931-f003:**
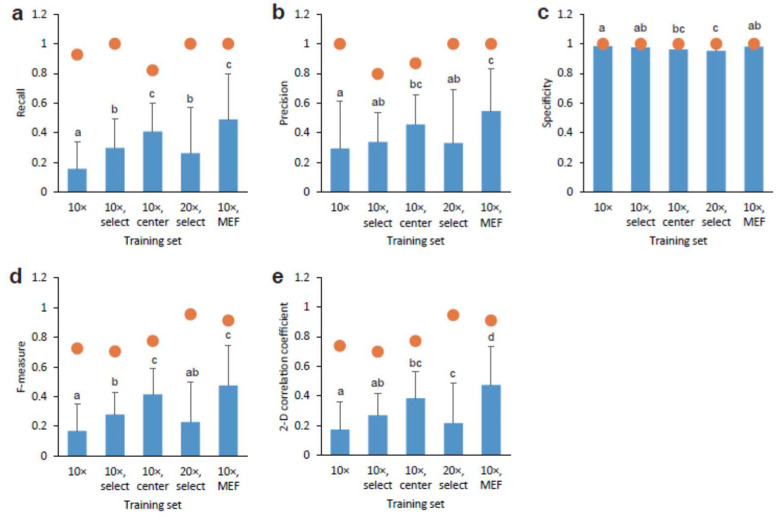
Comparison of depicted CSC images by AI models with original GFP fluorescence. Various AI models obtained from training sets were compared using AI output images and each true target image: (**a**) recall, (**b**) precision, (**c**) specificity, (**d**) F-measure, and (**e**) 2D correlation coefficient. Closed circles indicate maximum values. Mean ± S.D., *n* = 100 (exception: *n* = 40 for AI model obtained using training set 10× center). Identical letters labeled up the bars represent no significant difference, *p* < 0.05, and vice versa.

**Figure 4 biomolecules-10-00931-f004:**
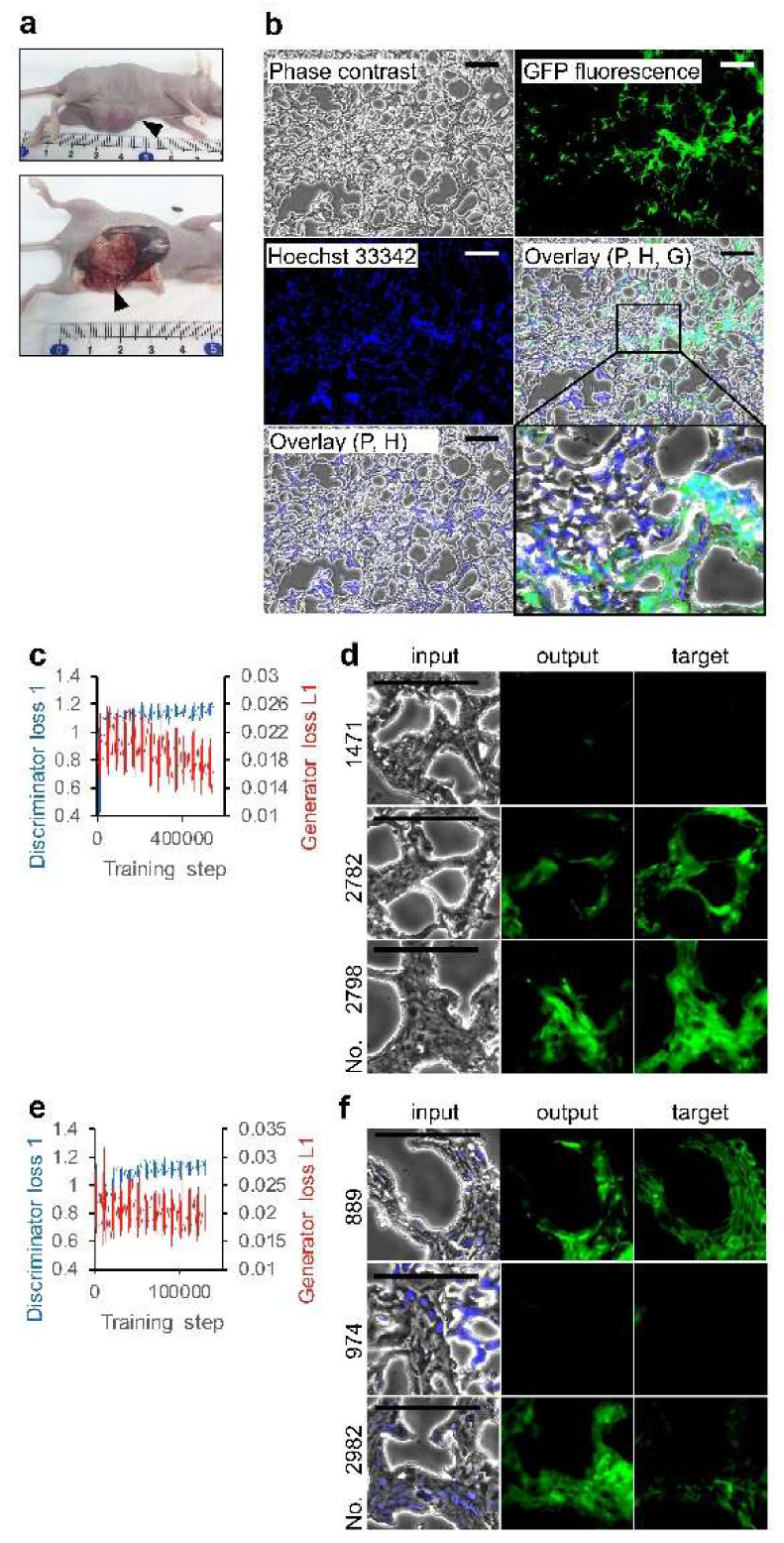
Deep learning of miPS-T47Dcm cell morphology in tumor tissue. (**a**) Primary subcutaneous tumors; arrowhead indicates tumor tissue. (**b**) Tumor tissue section visualized with phase contrast, Hoechst 33342, and GFP fluorescence using objection lens 20×. P: phase contrast; H: Hoechst 33342; G: GFP. Bars = 100 μm. An area in overlay (P, H, G) is shown in detail. (**c**,**e**) Effect of training steps on loss functions. (**d**,**f**) Output examples by AI models. Test phase contrast images were subjected to AI models for depicting fluorescence images. Input and target are the pair image for the depicted image evaluation. The AI models trained with the set of (**c**,**d**) phase contrast and GFP images, and (**e**,**f**) Hoechst 33342 overlaid-phase contrast and GFP images. Bars = 100 µm.

**Figure 5 biomolecules-10-00931-f005:**
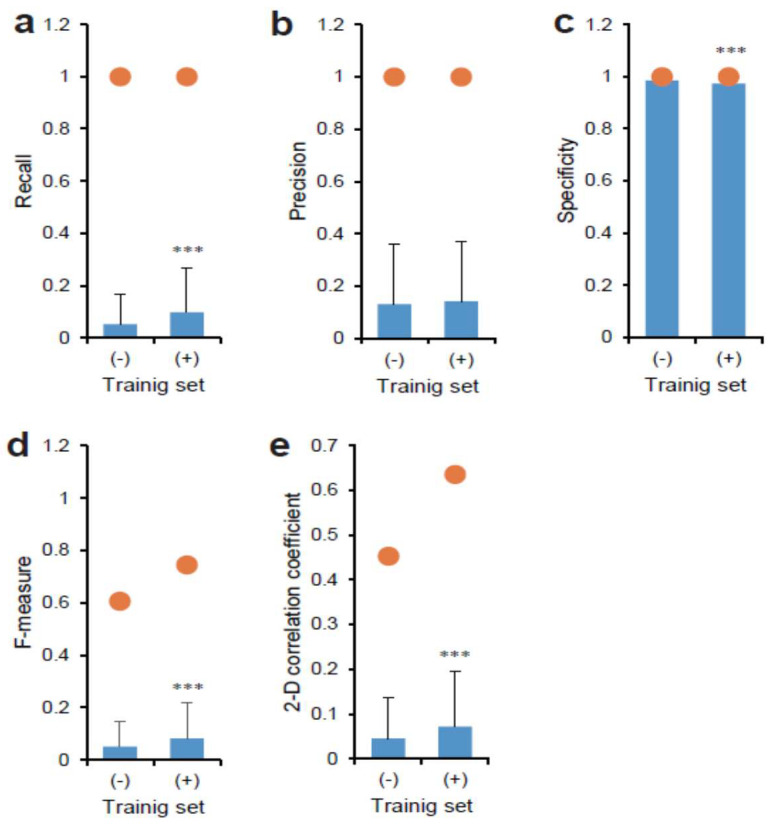
Comparison between depicted CSC image in tissue by AI models and original GFP fluorescence. Images of Hoechst 33342 overlaid-phase contrast (+) or not overlaid-phase contrast (-) were used for training AI for AI models. The AI output images and each true target image were compared using the values of (**a**) recall, (**b**) precision, (**c**) specificity, (**d**) F-measure, and (**e**) image correlation coefficient. Closed circles indicate maximum values. Mean ± S.D., *n* = 684. *** *p* < 0.01.

**Table 1 biomolecules-10-00931-t001:** Classification of cancer stem cells (CSCs) output images in tumor tissue

		**Set of Images for Training**			
		Phase contrast and GFP		Hoechst 33342 overlaid-phase contrast and GFP	
		**GFP Image Drawing in Output**			
		Yes	No	Yes	No
GFP fluorescence	Positive	95	341	129	296
	Negative	157	91	163	96

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
