# Peer review of "Deep Learning of Cancer Stem Cell Morphology Using Conditional Generative Adversarial Networks"

_biomolecules, 2020, doi:10.3390/biom10060931_

Round 1

Reviewer 1 Report

The manuscript title “Deep learning of cancer stem cell morphology using conditional generative adversarial networks” by Aida et al. focuses on training AI to identify cancer stem cell morphology in in vitro culture and in vivo tumor model. The manuscript sounds very exciting and should appeal to a broad audience. However, there are several key aspects of the manuscripts that need to be addressed first before publication.

Major:

  1. There are other acronyms that need to be defined first before using subsequently.
  2. Figure quality needs to be improved. The fonts are broken. They are not publication-ready.
  3. Some of the input images are overexposed. While acquiring of images, it is recommended to optimize intensity, exposure, etc. parameters throughout the experiment. This might affect your entire analysis.
  4. The accuracy of the AI is not high enough. In some cases, only n=40 for training sets were used, is it sufficient? It could trigger false positives/ negatives. How do you address the problem?
  5. I was expecting to see if the trained AI can detect CSC by morphological investigation alone which needs to be demonstrated.
  6. A negative control experiment is required where the trained AI demonstrates its aptitude.

Minor:

  1. A thorough language check is recommended. For example, if I understood the context in line 24 “remains elucidated” should be “remains elusive”.
  2. Line 63, 64, 71, etc.: what is the acronym “SCs”? It has not been defined earlier.

Reviewer 2 Report

In this manuscript, Aida et al demonstrated the feasible use of deep-learning workflows to map CSC morphology. Since there has been limited description of the morphology of CSCs in tumors, GFP fluorescence was used to define iPS-derived CSCs for training and mapping from phase-contrast images of CSCs. Following the process of image selection for training and nucleus staining, AI might recognize and distinguish the morphology of iPS-derived CSCs in phase-contrast tumor tissue images. Overall, the manuscript is well written with interesting data presented, which may provide insights into future AI-based applications for clinical guidance of cancer diagnosis and prognosis. This reviewer has no major concerns but a few minor points for the authors to consider to incorporate into revision.

  1. CSCs in tumors have been characterized by diverse stem cell markers. Just using Nanog may not be sufficient to reliably distinguish CSCs from other cancer cells.

  1. The GFP fluorescence emitted from cancer cells and tumor tissues showed large variations of signal intensity. How was positive staining defined?

  1. The center area of images was improved slightly compared to those without selection. How about the function values?

  1. The use of 20x objection did not reach a marked improvement. When using center images, did it prove?

  1. The use of MEF feeder cells showed the highest values of training sets. How about those with selection of eliminating blanks and the center?

  1. The definition and formula of specificity and F-measure should be explained.

  1. The data in Table 1 may need chi-square test to verify whether the increased ratio is statistically significant.

  1. In Figure 5c, the mean specificity values were almost 1.0 for both training sets, why p value still <0.01?

Round 2

Reviewer 1 Report

I am fine with this revised draft. Therefore, I recommend this manuscript for publication.